# Role of Açaí *(Euterpe oleracea)* in Modulating the Immune Response During Experimental Oral Infection with *Trypanosoma cruzi*

**DOI:** 10.3390/microorganisms13122711

**Published:** 2025-11-28

**Authors:** Flávia de Souza Marques, Thays Helena Chaves Duarte, Viviane Flores Xavier, Aline Coelho das Mercês, Thaís Vieira de Carvalho Silva, Luciana da Fonseca Medeiros, Camilo Elber Vital, Cláudia Martins Carneiro, William de Castro Borges, Paula Melo de Abreu Vieira

**Affiliations:** 1Laboratory of Morphopathology, Department of Biological Sciences, Nucleus of Biological Sciences Research, Institute of Exact and Biological Sciences, Federal University of Ouro Preto, Ouro Preto 35400-000, Brazil; flavia.marques@ufop.edu.br (F.d.S.M.); or thaysduarte.1993@gmail.com (T.H.C.D.); viviane.xavier@ufop.edu.br (V.F.X.); aline.merces@aluno.ufop.edu.br (A.C.d.M.); thais.vieira@aluno.ufop.edu.br (T.V.d.C.S.); 2Laboratory of Immunopathology, Nucleus of Biological Sciences Research, Institute of Exact and Biological Sciences, Federal University of Ouro Preto, Ouro Preto 35400-000, Brazil; luciana.medeiros@aluno.ufop.edu.br (L.d.F.M.); carneirocm@ufop.edu.br (C.M.C.); 3Enzymology and Proteomics Laboratory, Department of Biological Sciences, Nucleus of Biological Sciences Research, Institute of Exact and Biological Sciences, Federal University of Ouro Preto, Ouro Preto 35400-000, Brazil; camilo.vital@ufop.edu.br (C.E.V.); wborges@ufop.edu.br (W.d.C.B.); 4Department of Clinical Analysis, School of Pharmacy, Federal University of Ouro Preto, Ouro Preto 35400-000, Brazil

**Keywords:** Chagas disease, *Trypanosoma cruzi*, oral infection, açaí, stomach, proteomic

## Abstract

Oral infection is now the main route of Chagas disease transmission in endemic countries, with açaí (*Euterpe oleracea*) being the primary food involved in Brazil. However, the role of açaí in parasite–host interaction remains largely unexplored. This study assessed the effect of açaí inoculum on experimental *Trypanosoma cruzi* infection. BALB/c mice were orally infected with metacyclic trypomastigotes in RPMI or açaí. No difference in survival was observed. Tissue parasite load showed higher gastric parasitism in the RPMI group on day 5 after infection. Proteomic analysis of the açaí group revealed increased levels of cytoskeletal keratins and mucins, along with decreased pro-inflammatory cytokines and markers of tissue repair, indicating modulation of gastric inflammation. Both infected groups exhibited higher levels of gastrointestinal proteins (acid chitinase, gastrocin 1, trefoil) associated with mucosal protection and parasite clearance. These findings suggest that oral infection with açaí occurs more subtly, possibly due to decreased gastric inflammation, and highlight potential biomarkers for oral Chagas disease.

## 1. Introduction

Endemic in 21 Latin American countries, Chagas disease is caused by the protozoan *Trypanosoma cruzi* (*T. cruzi*) [1]. The World Health Organization (WHO) estimates that 6–7 million people are infected worldwide, with tens of thousands of new cases reported each year and over 10,000 deaths annually [2]. The primary transmission route in these areas is vectorial transmission, which occurs when trypomastigotes in the excreta of blood-sucking triatomines come into contact with broken skin or mucosa near the bite site [1,3]. However, following intensive health surveillance, vector transmission has decreased in endemic countries [4,5,6]. As vectorial transmission declined, other transmission routes—such as congenital transmission, blood transfusion, and organ transplantation—have become more relevant, especially in non-endemic countries. Still, in Latin American endemic regions like Brazil, Venezuela, and Colombia, oral transmission has recently emerged as the primary mode of *T. cruzi* spread [7,8,9,10,11]. Though oral infection has gained recognition mainly in recent decades, it is believed to be the natural transmission route in the sylvatic cycle, in which mammals acquire the disease by eating infected triatomines or by ingesting food contaminated with triatomine excreta [12,13]. Oral infections usually occur as micro-outbreaks, often affecting members of the same household. A notable feature of these outbreaks is the severity of illness: patients frequently experience high parasitemia, fever, and severe myocarditis, which can quickly lead to death during the acute phase [14]. This suggests that the oral route may trigger a different disease process than vectorial transmission.

In Brazil, various foods such as açaí, bacaba berries, and sugarcane juice have been linked to oral infection. In the North region—where vector control efforts have not been entirely successful—açaí appears to be the primary food involved in oral transmission. This Amazon-native fruit is highly energy-dense, providing lipids, proteins, and minerals [15]. As a result, açaí is part of the daily diet of local residents. It is also known that some triatomine species use the açaí palm tree for feeding or shelter. Additionally, in the North, açaí palm leaves are often used as a roofing material, which can facilitate insect domiciliation. Because açaí pulp extraction is usually performed without optimal hygiene practices, contamination can occur [13].

From a biological standpoint, oral infection poses challenges for both the host and the parasite. Parasites ingested with food must overcome physical and chemical barriers, including digestive enzymes, acidic gastric pH, and the protective mucus layer [16]. Studies indicate that orally infected mice exhibit stronger inflammatory responses and greater tissue damage than those infected via vector. However, the mechanisms behind this are still not well understood [17]. Moreover, the food itself might influence parasite viability and host-pathogen interactions. It has been shown that *T. cruzi* can survive in açaí adipose tissue even after freezing [18]. Furthermore, açaí contains bioactive compounds such as polyphenols and anthocyanins, known for their antioxidant and anti-inflammatory effects [19]. These properties raise the intriguing possibility that the fruit could modulate host immune responses during infection, potentially worsening or reducing tissue damage.

Although açaí is the main food associated with acute outbreaks in Brazil, no studies have examined how the environment in which the parasite is ingested affects its initial interaction with the vertebrate host. It is essential to determine whether components of açaí influence parasite survival in the stomach, alter local immune responses, or affect systemic disease outcomes. Filling this knowledge gap is vital for understanding the mechanism of oral transmission and identifying potential biomarkers and therapeutic targets. Additionally, insights into how the food matrix influences host–parasite interactions could inform other foodborne parasitic diseases.

In this context, the present study aimed to assess how açaí inoculum influences experimental *T. cruzi* infection in a mouse model. By comparing infection outcomes between parasites delivered in açaí versus standard culture medium, we aimed to determine whether the food matrix affects parasite establishment, tissue distribution, or host immune responses. This approach combines epidemiological observations with mechanistic insights, offering a new perspective on oral Chagas disease and deepening knowledge of this increasingly important transmission route.

## 2. Materials and Methods

### 2.1. Açaí

The açaí used in this study was obtained in August 2020 at the Ver-o-Peso market in Belém, Pará, Brazil. The fruits were transported frozen to Ouro Preto, Minas Gerais, Brazil, where they were stored at −20 °C. On the day of pulp extraction, the fruits were thawed and then subjected to a bleaching process to remove contaminants. The açaí pulp was produced by passing the fruits through a sieve to separate the pulp from the seeds. The nutritional characterization of açaí, along with the determination of its polyphenol content and antioxidant capacity, was conducted, and the results are available in the Appendix A.

### 2.2. Animals and Infection with T. cruzi

BALB/c mice, weighing approximately 20 g and aged 30 to 40 days, were housed at the Centro de Ciência Animal of the Universidade Federal de Ouro Preto (CCA-UFOP). For these experiments, only male mice were used because the estrous cycle in female mice could influence the immune response. All procedures adhered to the guidelines of the Conselho Nacional de Controle de Experimentação Animal (CONCEA). The project was approved by the Ethical Committee on Animal Research of the Universidade Federal de Ouro Preto (Approval ID: 6327160320).

On the day of infection, the mice in the infected groups fasted from food and water for 4 h before inoculation. They received 1 × 10^5^ metacyclic trypomastigotes from an acellular culture of T. cruzi, resuspended in RPMI medium or açaí, directly into the oral cavity with a gavage needle. The inoculum volume did not exceed 200 microliters.

### 2.3. Parasitemia Curve Assessment

The parasitemia curve was plotted using the methodology described by Brenner (1962) with some adaptations [20]. A total of 26 mice were monitored for 42 days post-infection (DPI). To summarize, the number of parasites was counted in 50 random microscopic fields from a 5-microliter wet blood preparation collected from the caudal vein of each animal.

### 2.4. Survival RateEvaluation

The animals used to establish the parasitemia curve were monitored daily until 42 DPI, and survival was recorded and expressed as a cumulative percentage.

### 2.5. Euthanasia and Tissue Sampling

BALB/c mice were euthanized through exsanguination via cardiac puncture (without thoracotomy) after being anesthetized with an intraperitoneal injection of Ketamine (90 mg/kg) and Xylazine (9 mg/kg) on days 2, 5, 7, and 14 post-infection. The stomach was collected and divided into two parts. The first part was preserved in a mixture of methanol and dimethyl sulfoxide (4:1, *v*/*v*) and embedded in paraffin for histological, proteomic analysis, and assessment of tissue parasitism. The second part was frozen at −80 °C for cytokine production analysis.

### 2.6. Parasite Burden

Fragments of stomach tissue preserved in methanol and dimethyl sulfoxide solution were processed using standard histological techniques and embedded in paraffin. Then, 200 μm-thick paraffin sections were obtained with a rotary microtome. The samples were placed in microtubes, and to remove the paraffin, 1.5 mL of xylene was added. The microtubes were centrifuged at 14,000 rpm for 3 min, and the xylene was discarded. This process was repeated. To completely remove the paraffin, 1.0 mL of 70% alcohol was added to the microtubes, which were then centrifuged under the same conditions. The supernatant was discarded, and to eliminate residual alcohol, the microtubes were placed in a sample concentrator at 1300 rpm and 37 °C for 10 min. DNA extraction was then performed using the WizardTM Genomic DNA Purification Kit (Promega, Madison, WI, USA) according to the manufacturer’s instructions.

### 2.7. Morphometric Analyses

The paraffin blocks described in the section Euthanasia and Tissue Sampling were cut using a rotary microtome to obtain 4-μm-thick sections. Tissue samples were de-waxed twice in xylene and hydrated in decreasing ethanol concentrations. The final step was washing in distilled water. The sections were then stained with hematoxylin and eosin. Images were captured using a Leica DM5000B microscope equipped with a camera and Leica Application Suite software (Version 2.4.0 R1). The images were analyzed using Leica QWin V3 software. The inflammatory process was quantified by counting nucleated cells in 30 random fields (total area 5493.7 µm^2^). All analyses were performed at the Multi-user Laboratory of the Núcleo de Pesquisas em Ciências Biológicas at the Universidade Federal de Ouro Preto (NUPEB-UFOP).

### 2.8. Cytokine Production Quantification

Cytokine production in the stomach was measured using the Cytometric Bead Array (CBA) Mouse Th1/Th2/Th17 Cytokine kit, following the manufacturer’s instructions. This kit enables the measurement of interleukin 2 (IL-2), interleukin 4 (IL-4), interleukin 6 (IL-6), interleukin 10 (IL-10), Tumor Necrosis Factor (TNF), Interferon-gamma (IFN-γ), and interleukin 17A (IL-17A). A 30 mg sample of stomach tissue, stored at −80 °C, was placed into microtubes with 500 μL of protease inhibitor cocktail (Sigma Chemical Co., St. Louis, USA). The tissue was then homogenized using a Tissuelyser (Qiagen). The microtubes were centrifuged at 10,000× g for 10 min at 4 °C, and the supernatant was collected and stored at −80 °C. For cytokine measurement, the supernatant was thawed in a 37 °C water bath, then centrifuged at 10,000× g for 10 min at room temperature. The supernatants were collected and processed according to the kit protocol.

### 2.9. Mass Spectrometry for Proteomics Analysis

Quantitative, label-free shotgun proteomic analysis was performed on triplicate stomach samples from non-infected and infected mice euthanized 5 days after infection.

### 2.10. In-Solution Digestion

In summary, the paraffin blocks described in the subheading Euthanasia and Tissue Sampling were sectioned on a rotary microtome into three 10 μm-thick sections, which were placed on glass slides. The sections were immediately de-waxed and then hydrated in solutions with decreasing ethanol concentrations, as described in [21]. The tissues were then placed in 1.5 µL microtubes and rinsed with 5–10 µL of ultrapure water in the shaved area. The samples were dried in a SpeedVac evaporator (RVC 2-18 CD Plus, Christ) for 120 min. For tryptic digestion of proteins, 30 μg of proteins were diluted in 160 μL of 25 mM Ammonium Bicarbonate. Next, 10 μL of 1% (*w*/*v*) RapiGest™ (Waters) was added to the samples, which were then incubated at 95 °C in a BIO-PLUS IT-2002 Incubator for 30 min. The proteins were reduced by incubating the samples with 9.2 mg/mL dithiothreitol (Sigma-Aldrich, St. Louis, MO, USA) at 60 °C for 10 min. The samples were then cooled to room temperature and alkylated with 33 mg/mL Iodoacetamide (Sigma-Aldrich) in the dark for 30 min. For digestion, sequencing-grade modified trypsin (Promega, Madison, WI, USA) was added at a protein: trypsin ratio of 50:1, and the samples were incubated at 37 °C for 12 h. To stop digestion and precipitate RapiGest™, 1% trifluoroacetic acid (Sigma-Aldrich) was added. The samples were then incubated again at 37 °C in a thermoblock for 30 min. After this period, the microtubes were centrifuged at 20,000× g for 15 min at 7 °C (Hettich Zentrifugen Mikro 200R, Tuttlingen, Germany). The supernatants were collected and transferred to vials for UHPLC analysis.

### 2.11. Liquid Chromatography-Mass Spectrometry

Following enzymatic digestion, approximately 500 ng of peptide samples were injected into the liquid chromatography system (UHPLC UltiMate R 3000, Dionex, San Jose, CA, USA). The peptides were directed to an Acclaim PepMap100 C18 Nano-Trap column (100 µm i.d. × 2 cm, 5 µm, 100 Å; Thermo Scientific, Waltham, MA, USA) and washed for 3 min in a 3.8% acetonitrile/0.1% trifluoroacetic acid solution (ACN, HPLC grade, USA) at a flow rate of 5 µL/min before proceeding to reverse-phase chromatography with an Acclaim PepMap100 C18 RSLC column (75 µm i.d. × 15 cm, 2 µm, 100 Å; Thermo Scientific). A multi-step gradient was applied using solvents A (0.1% formic acid, HPLC grade, JTBaker, Mexico) and B (80% ACN, 0.1% formic acid): starting with 3.8% B for 3 min, then increasing from 3.8% to 30% B over 120 min, followed by a rise from 30% to 55% B until 150 min. Finally, the gradient increased to 99% B for 162 min, then reconditioned with 3.8% B for 18 min. Spectral data were acquired using a Q-Exactive mass spectrometer (Thermo Scientific, Bremen, Germany) operated in full-scan/MS2 mode. The nanospray flex ion source (Thermo Scientific) was set to 3.8 kV in positive mode, with a capillary temperature of 250 °C and an S-lens level of 55. Data-dependent acquisition (DDA) targeted the top 12 ions with charge states ranging from +2 to +4 within a 1.2 *m*/*z* window. Survey scans were obtained at a resolution of 70,000 with a mass range of 300–2000 *m*/*z*, an AGC target of 1 × 10^6^ ions, and a maximum ion injection time of 120 ms. For MS/MS scans, selected ions were fragmented by higher energy collision dissociation (HCD) with a stepped normalized collision energy (NCE) of 28–30. Fragmentation spectra were acquired at a resolution of 17,500, with a maximum injection time of 60 ms, a dynamic exclusion time of 40 s, and a target value of 5 × 10^5^ (minimum AGC target 6.25 × 10^3^).

### 2.12. Analysis of Proteomic Data

The spectral data acquired from the UHPLC-MS/MS platform were used for protein identification with PEAKS Studio v8.5 (Bioinformatics Solutions Inc., Waterloo, ON, Canada). The search parameters included: enzyme: trypsin; up to 2 missed cleavages; fixed post-translational modifications: cysteine carbamidomethylation (+57.0214 Da) and methionine oxidation (+15.9949 Da) as variable modifications, with no more than three modifications per peptide; time window: 4 min; mass error tolerance of 10 ppm for precursor ions and 0.1 Da for product ion fragmentation. The database search was conducted with a False Discovery Rate (FDR) of 1% and a protein significance threshold of 13 (*p*-value ≤ 0.05), and only proteins identified with at least one unique peptide were considered. The Mus musculus proteome database, downloaded from UNIPROT, contained 55,398 protein sequences. Quantitative label-free analysis was performed with a fold-change ≥ 2 and protein significance of 13 (*p*-value ≤ 0.05) using the PEAKSQ significance method, considering proteins identified with at least one unique peptide. Cytoscape Software v. 3.9.1 (available at https://cytoscape.org/) and the STRING query tool were used to construct a protein interaction network for differentially abundant serum proteins and to identify the top 10 most significant enriched molecular functions associated with them.

### 2.13. Statistical Analysis

Statistical tests were conducted using GraphPad Prism 5 (Prism Software, Irvine, CA, USA). To evaluate normality, data were analyzed with the Kolmogorov–Smirnov test. For parametric data, a *t*-test (only for infected groups) or one-way analysis of variance (ANOVA) (for both non-infected and infected groups) was employed. When significant differences were identified, post hoc analysis was conducted using the Bonferroni test. For non-parametric data, Mann–Whitney tests (for comparisons among infected groups only) or Kruskal–Wallis tests followed by Dunn’s test were used. Differences between means were considered statistically significant when *p*-values were <0.05.

## 3. Results

### 3.1. Survival Rate Outcomes 

No deaths occurred in the uninfected group, and there were no statistical differences in survival among any of the three groups. Mice in the RPMI group had a survival rate of 83%, while mice in the Açaí group had a survival rate of 98% (Figure 1).

### 3.2. Parasitemia Curve Results

The parasitemia curve is shown in Figure 2. The prepatent and patent periods for animals infected orally with RPMI media were 4 and 26 days, respectively. The peak of parasitemia occurred on the 12th day after infection, with 13,108 trypomastigotes per 0.1 mL of blood. In the Açaí group, the prepatent and patent periods were 11 and 29 days, respectively. Parasitemia peaked on the 16th day after infection, with 11,475 trypomastigotes per 0.1 mL of blood. Parasitemic values are summarized in Table 1.

### 3.3. Parasitic Load

To examine the effect of infection on tissue parasitism, the parasitic load in the stomachs of mice infected with the Y strain of *T. cruzi* in RPMI and Açaí media was measured. A higher parasitic load was seen in the stomachs of mice infected with RPMI media on the fifth day after infection compared to mice infected with Açaí (Figure 3).

### 3.4. Inflammatory Process

The results of the quantitative analysis of the inflammatory process are shown in Figure 4 and Figure 5. The morphometric analysis of the stomach’s muscular layer indicated that the infected groups exhibited significantly increased inflammatory responses on days 2 and 5 after infection compared to the control group.

### 3.5. Cytokine Production

The results regarding the production of pro-inflammatory muscle cytokines and the immunomodulatory cytokine IL-10 in the stomach are shown in Figure 6. Overall, animals infected with metacyclic trypomastigote forms in RPMI medium showed an increase in the production of both pro-inflammatory cytokines (IFN-γ, TNF) and the immunoregulatory cytokine IL-10 compared to animals infected with metacyclic forms in Açaí medium on the 2nd and 7th days after infection. In contrast, a reduction in cytokine production was generally observed in the Açaí group compared to the control group. This reduction was noted at all evaluation time points for the pro-inflammatory cytokine TNF and the cytokine IL-17. A decrease in the pro-inflammatory cytokine IFN-γ was observed on the 7th day after infection. Conversely, reductions in the immunoregulatory cytokine IL-10 were seen on both the 2nd and 7th days after infection.

### 3.6. Stomach Proteomic Analysis Based on Mass Spectrometry

#### 3.6.1. Compositional Analysis

For large-scale proteomic analysis, triplicate stomach samples from control and infected animals euthanized 5 days after infection were used. Indicators obtained from the UHPLC-MS/MS platform are shown in Table 2. An FDR cutoff of 1.1% for spectra, 4.3% for peptides, and 5.3% for protein identification was applied. A total of 1887 proteins were identified, resulting in 1251 protein groups (Appendix A) shared across all experimental groups.

Of the 1251 proteins identified, the majority 781 (64.3%) were shared across the three experimental conditions (Figure 7). The number of unique proteins identified in the control group was 92 (7.6%). In the RPMI group, this number was 35 (2.9%), and in the Açaí group, 47 proteins (3.9%) were exclusive to that group. The list of proteins unique to each condition can be found in Appendix A.

The most common proteins shared among the three experimental groups mainly participate in RNA processing (splicing), the respiratory chain, protein synthesis, the proteasome, amino acid breakdown, and cell formation. The biological pathways related to these proteins are shown in Figure 8.

The cumulative abundance plot (Figure 9A) displays the contribution of each protein to the total ion signal in the stomach proteome. Only 122 proteins accounted for 50% of the ion signal. Additionally, proteins with the most minor contribution to the ionic signal made up 10% of the extract, corresponding to 650 proteins. To further analyze their contribution to the total ion signal, a log10 plot of spectral counts was generated for each protein, arranged according to their contribution to the total ion signal. This graph shows a detection range spanning four orders of magnitude, from the most abundant protein to the least. (Figure 9B).

Figure 10 shows that the 122 proteins are involved in pathways mainly related to protecting the epithelial barrier, muscle contraction, DNA compaction and decompression, protein regulation, and the respiratory chain.

#### 3.6.2. Quantitative Analysis

The quantitative analysis of the stomach proteome was conducted using label-free shotgun proteomics with fold change criteria of ≥2 and significance threshold of ≥20. This analysis showed that, of the 1251 proteins identified during compositional analysis, 110 (8.8%) were differentially abundant (Figure 11). However, 49 proteins lacked a measurable ionic signal area and were excluded from further analysis. Of the remaining 61 proteins, 24 (39.3%) were upregulated in the Açaí group and downregulated in the RPMI group, both relative to the control group. In the infected groups, 17 proteins (27.8%) were downregulated, and 17 (27.8%) were upregulated, compared to the control group. Only 3 proteins were positively regulated in the RPMI group and negatively regulated in the Açaí group, relative to the control group. Data on all differentially abundant proteins with quantifiable ionic signal areas are listed in Appendix A.

## 4. Discussion

In recent decades, oral infection has become the main route of transmission for Chagas disease in endemic countries. In Brazil, the northern region accounts for the highest number of reported cases and deaths [22]. In this area, açaí, a native fruit from the Amazon, is the food most often associated with oral outbreaks [13]. Açaí pulp is already recognized as a functional food because of its chemical makeup and health-related bioactivities [19], mainly due to phenolic compounds that have known antioxidant and anti-inflammatory properties [23,24]. However, there are no studies in the literature examining the environment in which the parasite interacts with the vertebrate host or the impact of this initial interaction on the progression of the infection. Therefore, this study is the first to demonstrate how the inoculum medium can affect the immune response, tissue damage, and protein production in experimental oral infection with Chagas disease.

Among the few studies in the literature that examine experimental oral infection with *T. cruzi*, there is no agreement on the best method for inoculum administration—such as incomplete or oropharyngeal gavage, intragastric gavage, or direct inoculation into the oral cavity [25,26,27]. Additionally, factors such as the strain used, inoculum size and volume, and the route of inoculation influence the course of infection, affecting parameters such as survival rate and the parasitemia curve. In human diseases, the higher mortality rate seen in acute symptomatic cases (8–35%) [28] may be linked to the number of parasites ingested during oral outbreaks, as a single triatomine can release over 600,000 trypomastigote forms. In contrast, the number of trypomastigotes in feces from vector transmission is roughly 4000 forms per microliter [13].

In this study, a mortality rate of 17% was observed in the RPMI group, while only 2% was seen in the Açaí group. In a previous survey [17] the mortality rate was 85% for BALB/c mice infected directly in the oral cavity with an inoculum half the size used in this study and a strain from a different DTU (VI) [17]. Another investigation [27] showed a mortality rate of 20%, similar to that in the RPMI group. Although this study used a Peruvian strain from the same DTU (II) as the strain in this work, the experimental variables differed: Swiss mice were used, the inoculation route was intragastric, and blood trypomastigotes were used for infection. The differences across these studies highlight the need to standardize these parameters for experimental oral *T. cruzi* infection.

In our study, although the parasitemia curves were similar, animals in the Açaí group took longer to detect parasites in peripheral blood (11 days), suggesting a more subtle infection. In another study, Swiss mice were intraperitoneally with metacyclic trypomastigote forms of the Y strain of *T. cruzi* [29]. They observed a peak parasitemia of 568,800 parasites per 0.1 mL of blood, a higher value than ours, even though their inoculum was 20 times smaller. Additionally, studies have shown that parenteral (intravenous, subcutaneous, and intramuscular) routes of infection are associated with higher levels of parasitemia, disease, and mortality than mucosal (oral, intragastric, intrarectal, conjunctival, or genital) routes [30,31,32,33]. In oral infection, a more extended pre-patent period and a peak parasitemia with fewer circulating forms may be due to the physical and chemical barriers the parasite encounters as it enters the bloodstream [16].

Given the lack of agreement in the literature on how oral inoculation is performed, there has been significant debate about the parasite’s entry point into the vertebrate host. Moreover, despite a higher burden, stomach infection in these animals was only established on the 14th day post-infection (DPI). This finding supports previous observations showing that *T. cruzi* can invade and replicate in the gastric mucosal epithelium but not in the oropharyngeal or esophageal mucosa [26].

It is well known that effective control of *T. cruzi* infection depends on a balance between pro-inflammatory cytokines, which promote parasite destruction, and anti-inflammatory cytokines, which reduce tissue damage. In both infected groups, regulation of the inflammatory response was observed during the later stages of this study. This may be related to the production of the cytokine IL-17A, which helps regulate *T. cruzi* infection by recruiting and activating IL-10-producing neutrophils, thus limiting tissue damage and mortality [34,35]. Additionally, an increase in IL-17 production has been previously observed in mucosal *T. cruzi* infection [17]. IL-17 is also known to contribute to forming a gastrointestinal barrier by activating mucosal defenses to eliminate pathogens [36].

Proteomic analysis offered insights into the dynamics of tissue parasitism and the inflammatory response in the animals’ stomachs. Notably, the protein abundance profile in the Açaí group more closely resembled that of the control group than that of the RPMI group. Among the proteins identified in the Açaí group, we highlight cytoskeletal keratins and mucin 16. Keratins are intermediate filament proteins in epithelial cells, and some studies have shown they play a role in maintaining the mechanical stability and integrity of epithelial cells [37]. Additionally, specific keratins contribute to immune response and wound healing by promoting macrophage polarization toward the M2 phenotype, increasing the production of the immunoregulatory cytokine IL-10 and reducing pro-inflammatory cytokines like IL-1β and IL-6 [38,39]. Mucin 16 helps downregulate IL-6 production, favoring wound healing. Together, these data may explain the control of the inflammatory process in the Açaí group after the 5th day of infection.

The life cycle of *T. cruzi* involves an obligatory intracellular phase in the vertebrate host, and efficient invasion requires the rearrangement of the host’s cytoskeletal components [40,41,42]. Trypomastigote forms bind to various extracellular matrix components to reach host cells [43]. In the Açaí group, increased protein levels related to cytoskeleton organization and extracellular matrix remodeling, as well as negative regulation of cell adhesion (e.g., Tubulin β-2, Hemartin, Fibulin), were observed. Studies have shown that pharmacological inhibition of microtubule dynamics reduces parasitic invasion into non-phagocytic cells such as fibroblasts and myocytes [44].

Both Açaí- and RPMI-infected groups shared 57 proteins, including LIM and SH3 domain protein 1, Galectin-2, Mammalian acid chitinase (CHIA), Trefoil factor 1 (TFF1), and Gastrocin-1 (GKN1). LIM and SH3 domain protein 1 accumulates in focal adhesions, facilitating cell adhesion, migration, and communication by binding to actin filaments [45]. An increase in this protein in infected groups may be associated with *T. cruzi* internalization.

Galectin-2 is highly expressed in the gastrointestinal tract and plays a role in regulating physiological and pathological processes, including maintaining epithelial integrity, regulating inflammatory and immune responses, and promoting apoptosis [46]. It binds to glycoproteins on T cells, increasing IL-10 and IFN-γ production, leading to T cell apoptosis [47]. In the RPMI group, increased production of these cytokines may have contributed to a higher inflammatory response, leading to tissue damage and increased mortality.

CHIA is an enzyme present in the lungs and gastrointestinal tract of both humans and mice. Research has shown that this enzyme increases in epithelial cells and macrophages during gastrointestinal helminth infections, providing a protective role by steering the inflammatory response toward the Th2 subtype [48]. In our study, we observed an increase in this enzyme in the stomachs of animals across both infected groups. This rise may be part of the host’s protective mechanism against *T. cruzi* infection.

The TFF1 protein is mainly expressed in the gastric epithelium and helps preserve mucosal integrity [49]. However, some research has shown a connection between inflammatory bowel diseases, such as chronic pancreatitis, Barrett’s esophagus, peptic ulcers, and inflammatory bowel disease, and increased TFF1 production through mechanisms that are not yet fully understood [50]. In our study, TFF1 levels rose in both infected groups but not in the control group. This rise may have resulted from tissue damage caused by the body’s efforts to eliminate *T. cruzi*, aimed at maintaining the integrity of the gastrointestinal epithelium.

In both the RPMI and Açaí-infected groups, a higher abundance of the stomach-specific protein GKN1 was observed. Some studies suggest that GKN1 has a role in repairing gastric mucosa after tissue injury by promoting cell proliferation and differentiation. Previous work demonstrated that applying GKN1 to gastrointestinal cells restored the epithelium in mice with colonic epithelial injury [51,52]. In this context, the increase in this protein in our study may indicate an effort to restore gastric mucosal integrity after tissue damage caused by *T. cruzi*. Since there was no increase in the abundance of this protein in control animals, GKN1 could serve as a potential biomarker for oral Chagas disease.

## 5. Conclusions

Given these results, we can conclude that in experimental oral infection by *T. cruzi* with Açaí, the infection occurs more silently with a longer pre-patent period of parasitemia. Furthermore, the infection appears in a subclinical form, with the establishment of infection in the stomach only on the 14th day after infection, along with a reduction in pro-inflammatory cytokine production. Proteomic data support these findings, as a protein profile similar to that of the control group observed in the stomachs of animals infected with Açaí.

## Figures and Tables

**Figure 1 microorganisms-13-02711-f001:**
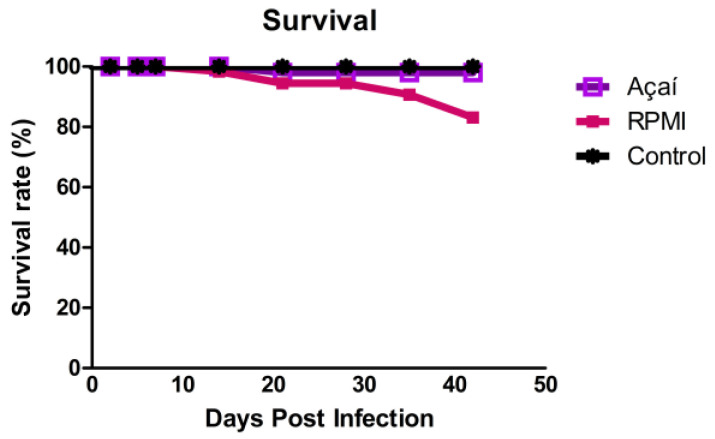
Survival rate of BALB/c mice uninfected (■) or infected with metacyclic trypomastigote forms of the Y strain of *T. cruzi* in RPMI (■) or Açaí (■). Each curve shows the average of 26 animals from each group infected over 42 days post-infection.

**Figure 2 microorganisms-13-02711-f002:**
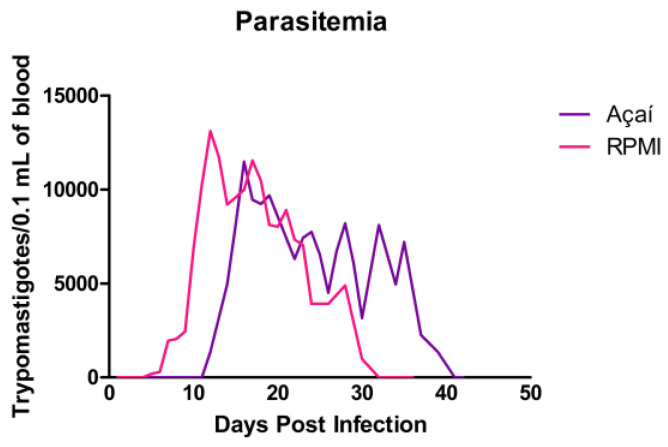
Parasitemia in BALB/c mice infected with metacyclic trypomastigotes of the Y strain of *T. cruzi* in RPMI (■) or Açaí (■). Each curve shows the average of 26 animals from each infected group, measured 42 days after infection.

**Figure 3 microorganisms-13-02711-f003:**
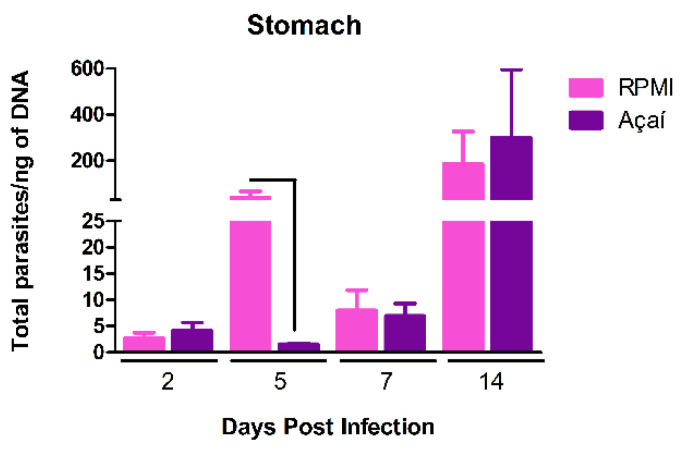
Number of parasites in the stomachs of BALB/c mice infected with metacyclic forms of the Y strain of *T. cruzi* in RPMI (■) or Açaí (■), euthanized during the acute phases (2, 5, 7, and 14 DAI). Results are shown as mean ± standard error. Connecting lines indicate a significant difference between groups (*p* < 0.05).

**Figure 4 microorganisms-13-02711-f004:**
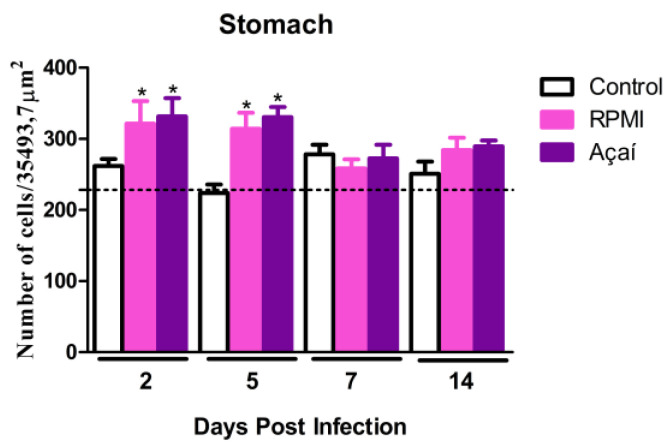
Quantification of the inflammatory process in the stomach of non-infected BALB/c mice (□), infected with the metacyclic trypomastigote forms of the Y strain of *T. cruzi* in RPMI (■) or Açaí (■). Values were expressed as mean ± standard error. “*” represents a significant difference between infected groups and the control group.

**Figure 5 microorganisms-13-02711-f005:**
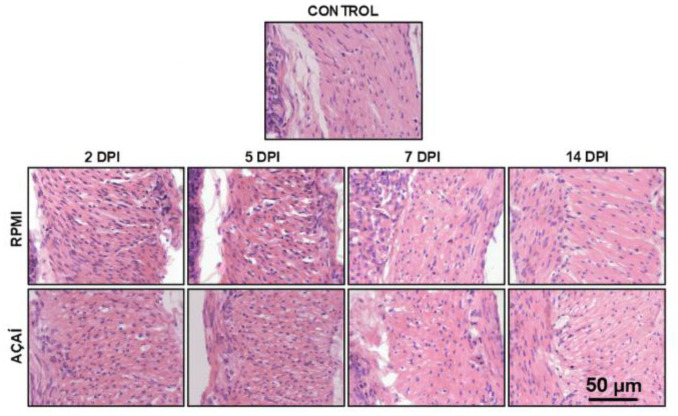
Photomicrographs of histological sections of the stomach’s muscular layer in BALB/c mice after infection with metacyclic trypomastigote forms of the Y strain of *Trypanosoma cruzi* in RPMI or Açaí. Normal muscle histology in non-infected animals (Control). Moderate inflammatory infiltrate observed two and five days after infection in both infected groups. Normal muscle histology seen in all groups on days seven and 14 after infection. Hematoxylin-Eosin stain. Bar = 50 μm.

**Figure 6 microorganisms-13-02711-f006:**
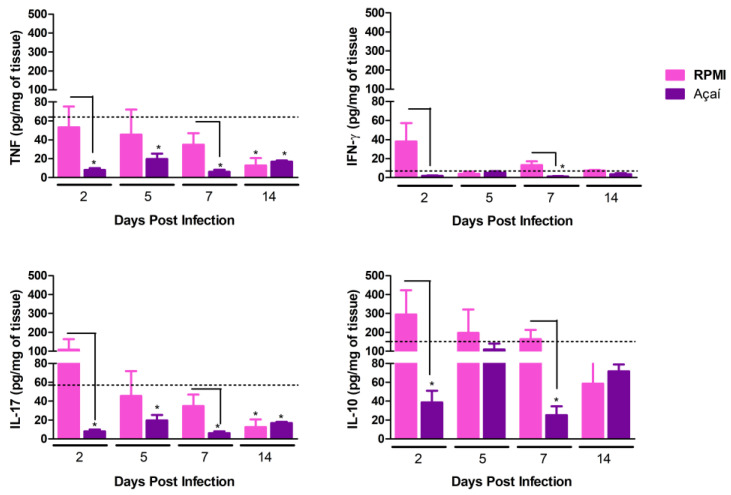
Quantification of cytokines in the stomach of uninfected BALB/c mice (dashed line) and infected BALB/c mice with metacyclic trypomastigote forms in RPMI (■) or Açaí (■) of Y strain. “*” indicates a significant difference between infected groups and the control group. Connecting lines show significant differences over time between groups of infected animals. Values are expressed as mean ± standard error of six mice per group at each time point.

**Figure 7 microorganisms-13-02711-f007:**
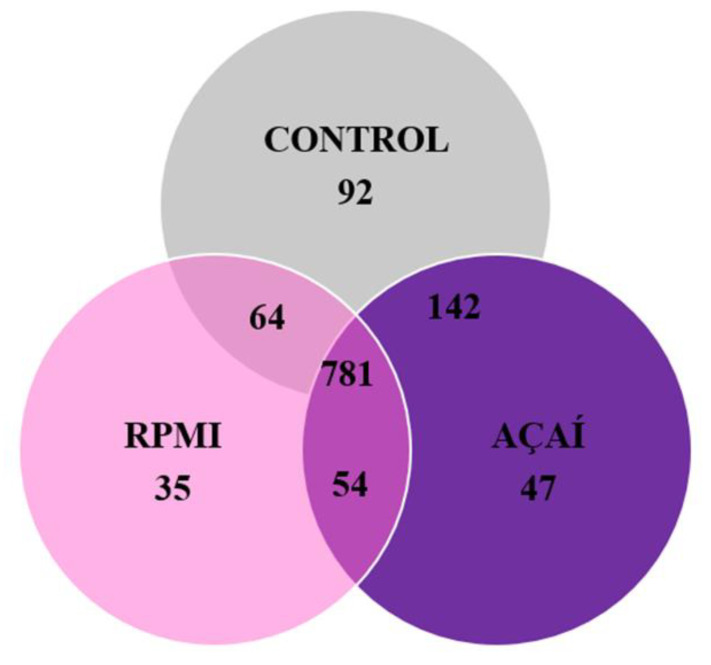
Venn diagram showing the shared proteins among the three experimental groups.

**Figure 8 microorganisms-13-02711-f008:**
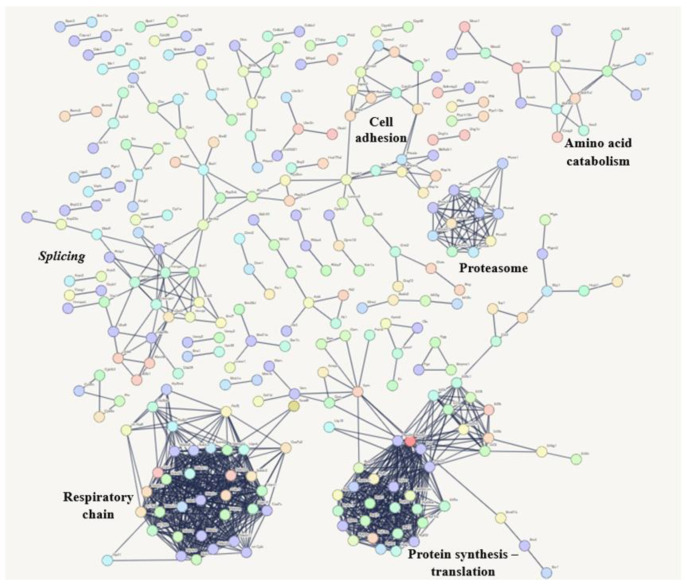
The interaction network of proteins involved in biological processes shared by the three experimental conditions. The network was generated using String Version 12.0, based on the *Mus musculus* database. A confidence cutoff score was set at 0.9 or higher.

**Figure 9 microorganisms-13-02711-f009:**
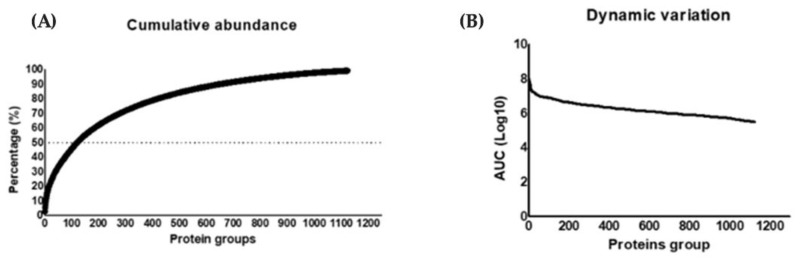
(**A**) Cumulative abundance: distribution of protein abundance in the stomach proteome. (**B**) Dynamic variation on a logarithmic scale (log_10_), showing detection across four orders of magnitude by the UHPLC-MS/MS platform.

**Figure 10 microorganisms-13-02711-f010:**
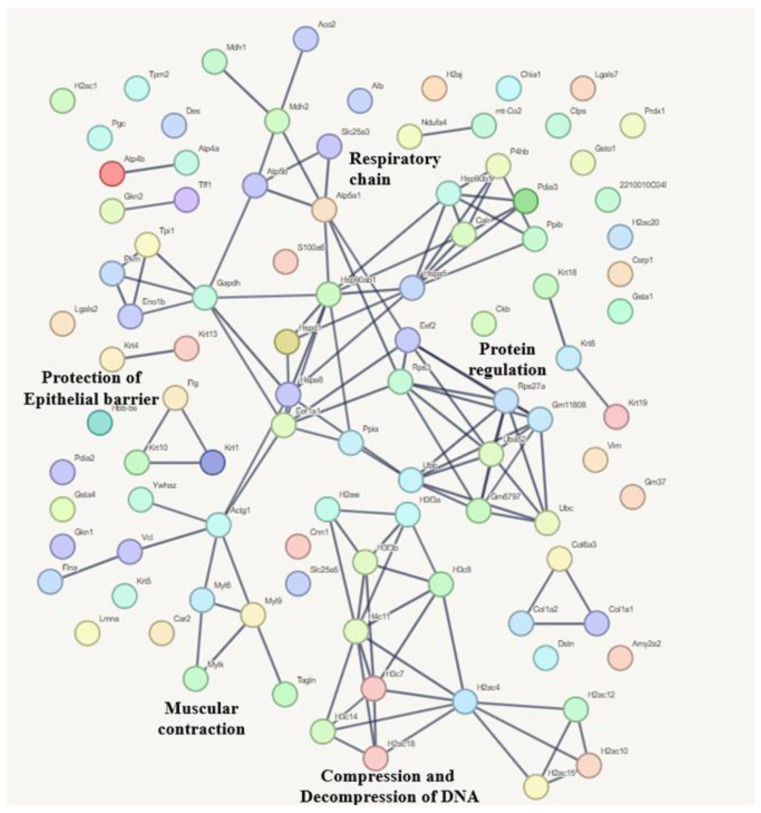
The interaction network of proteins involved in biological processes, in which the proteins contributing to 50% of the ionic signal are involved. The network was generated using the STRING version 12.0 program, based on a Mus musculus database. The selected confidence cutoff score was >0.9.

**Figure 11 microorganisms-13-02711-f011:**
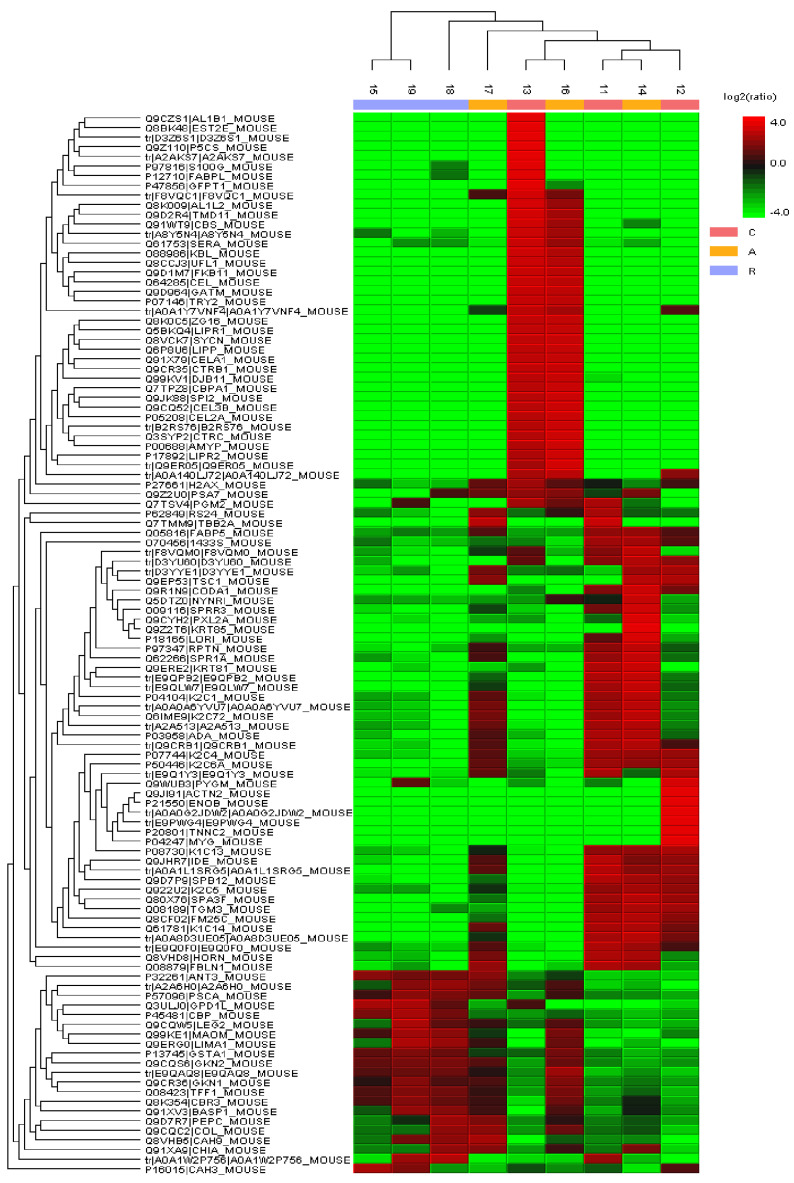
Heatmap showing the differential expression profile of the 110 proteins identified in the quantitative analysis. Green rectangles indicate downregulated proteins, while red rectangles indicate upregulated proteins.

**Table 1 microorganisms-13-02711-t001:** Parasitemic values referring to animals infected with metacyclic trypomastigote forms of the Y strain of *T. cruzi* in RPMI medium or Açaí.

Groups	PPP	PP	MPP	DMPP
RPMI	4	26	13,108	12
Açaí	11	29	11,475	16

PPP: pre-patent period; PP: patent period; MPP: value of trypomastigotes at the maximum peak of parasitemia (trypomastigotes/0.1 mL of blood); DMPP: day of the maximum peak of parasitemia).

**Table 2 microorganisms-13-02711-t002:** Indicators obtained from the UHPLC-MS/MS platform covering all groups.

Indicators	Results
MS	242.783
MS/MS	178.876
Peptide-Spectrum Matches (PSM)	52.802
Peptide sequences	7.760
Protein groups	1251
Proteins (unique peptides)	1012 (>2); 369 (=2); 506 (=1)

## Data Availability

The original contributions presented in this study are included in the article/Appendix A. Further inquiries can be directed to the corresponding author.

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
