# Peer review of "Role of Açaí (Euterpe oleracea) in Modulating the Immune Response During Experimental Oral Infection with Trypanosoma cruzi"

_microorganisms, 2025, doi:10.3390/microorganisms13122711_

Round 1
Reviewer 1 Report
Comments and Suggestions for Authors
- A very clever study, the authors are to be commended. I wondered after studying Fig. 1 if waiting until the mice died or at least a longer time period, whether there might be a significant difference in controls and Acai group. What is the natural history of the control? Do they slowly become indeterminate or do they all die?
- I suggest striking line 112-113 where you mention what might be done in future trials.
- Are there any proteins detected that actually suggest some pattern related to infection or is that we are to conclude there is a significant difference in shared proteins among the three groups?
Author Response
Comment 1: A very clever study, the authors are to be commended. I wondered after studying Fig. 1 if waiting until the mice died or at least a longer time period, whether there might be a significant difference in controls and Acai group. What is the natural history of the control? Do they slowly become indeterminate or do they all die?
Response 1: Reviewer 1, we sincerely appreciate your thoughtful and insightful question about the natural history of the control group. Your comment helped us clarify an important aspect of our experimental design, and we are grateful for the opportunity to provide a more detailed explanation. Additionally, as requested by Reviewer 3, we have now included the survival data of the control group in Section 3.1 to improve the clarity and completeness of the results.
For this experiment, we focused solely on the acute phase of the disease, as mortality in human oral outbreaks typically occurs during this period. In our experimental model, the acute phase lasted approximately 40 days, as shown by the parasitemia curve. The uninfected control group showed no deaths during the 42 days of monitoring. At this stage, we do not yet know the long-term effects of oral Açaí infection in mice, as this experiment was not designed to extend into the chronic phase. However, we fully acknowledge the importance of this question and plan to conduct longer studies in the future to evaluate chronic outcomes.
Comment 2: I suggest striking line 112-113 where you mention what might be done in future trials.
Response 2: We sincerely appreciate your careful reading and helpful suggestion. As you recommended, we have removed lines 112–113 that mentioned potential future trials.
Comment 3: Are there any proteins detected that actually suggest some pattern related to infection or is that we are to conclude there is a significant difference in shared proteins among the three groups?
Response 3: We appreciate this insightful question. As noted in the Discussion section (lines 554–589), both infected groups (Açaí and RPMI) shared 57 proteins not found in the uninfected controls. Among these are proteins such as LIM and SH3 domain protein 1, Galectin-2, Mammalian acid chitinase, Trefoil factor 1, and Gastrocin-1. Many of these proteins are typical of the gastrointestinal tract, suggesting that the stomach may serve as the parasite's entry point.
Additionally, many of these proteins are involved in cytoskeletal remodeling, a process that may be linked to internalization. Others are related to tissue repair and maintaining mucosal integrity after injury, which aligns with the damage expected during infection. These findings suggest that the proteins uniquely upregulated in the infected groups may reflect biological processes directly related to infection and tissue response. We believe these proteins could be promising biomarkers for orally acquired Chagas disease. Next, we plan to evaluate serum from infected animals to determine whether any of these proteins or related pathways can be detected systemically.
Reviewer 2 Report
Comments and Suggestions for Authors
The MS explored the açaí in modulating the immune response of during experimental oral infection with Trypanosoma cruzi. The following aspects can be applied to improve the quarlity of the MS.
- The composition of pulp extraction from açaí needs to be tested to know main ingredients? And to further analyze which ingredient contribute the main function?
- As to Euthanasia and Tissue Sampling, how to confirm the concentration in treating animals?
- The scale is needed in Figure 5.
- There are quite a few format issues in the MS. Please check and revise them carefully.
Author Response
Comment 1: The composition of pulp extraction from açaí needs to be tested to know main ingredients? And to further analyze which ingredient contribute the main function?
Response 1: This is a critical point. We have characterized açaí pulp, and the results will be included as supplementary material. Specifically, we quantified its elemental composition, total polyphenol content, and antioxidant activity using the DPPH· and ABTS•⁺ free radical methods. In the Materials and Methods section (highlighted in green, lines 105–108), we specify that these analyses were conducted and that the results are available in the supplementary material.
Comment 2: As to Euthanasia and Tissue Sampling, how to confirm the concentration in treating animals?
Response 2: We appreciate the reviewer's critical question. Just before infection, the number of metacyclic trypomastigotes in the LIT medium was counted using a Neubauer chamber to verify the parasite concentration. After centrifugation, the parasites were resuspended in Açaí or RPMI and thoroughly homogenized before being injected into each animal. The inoculum volume was then adjusted to ensure each mouse received exactly 100,000 parasites in 200 µL. This process enabled us to confirm and standardize the concentration administered to the animals accurately.
Comment 3: The scale is needed in Figure 5
Response 3: We appreciate the reviewer's observation. The scale bar has now been added to figure 5 as requested.
Comment 4: There are quite a few format issues in the MS. Please check and revise them carefully
Response 4: We appreciate the reviewer's attention to this point. The manuscript will be carefully reviewed, and all formatting issues will be corrected throughout; a thoroughly revised version with proper formatting will be submitted.
Reviewer 3 Report
Comments and Suggestions for Authors
Souza Marques et al evaluated the influence of açaí inoculum on experimental Trypanosoma cruzi infection. They found that oral in-fection with açaí occurs more silently, possibly due to reduced gastric inflammation, and highlight potential biomarkers for oral Chagas disease. My only comments is:
- All the 3.1 Survival Rate, 3.2 Parasitemia Curve , 3.3 Parasitic Load and 3.5 Cytokine Production should be included control, did controls have dead?
Author Response
Comment 1: All the 3.1 Survival Rate, 3.2 Parasitemia Curve , 3.3 Parasitic Load and 3.5 Cytokine Production should be included control, did controls have dead?
Response 1: Reviewer 3, we sincerely appreciate your careful review of our manuscript and thank you for emphasizing the importance of clearly presenting the control groups throughout the Results section. Regarding Section 3.1 – Survival Rate, the control group is now clearly included in both the text (highlighted in green, lines 265 and 267) and the graph (indicated by the black line). For Sections 3.2 – Parasitemia Curve and 3.3 – Parasitic Load, we kindly request clarification that the control group is intentionally absent, as these animals are not infected with T. cruzi. Consequently, parasites are neither detected in the bloodstream nor in the tissues of these control animals, making their inclusion in these analyses inappropriate. In Section 3.5 – Cytokine Production, the control group (uninfected animals) is shown as a dashed line on the graph. This dashed line represents the average cytokine levels in healthy, uninfected animals, serving as a reference for normal cytokine production under non-infectious conditions. Values above this line indicate increased cytokine production (hyperproduction), while values below it represent decreased production relative to the normal baseline.